# A Review on the Immunological Response against *Trypanosoma cruzi*

**DOI:** 10.3390/pathogens12020282

**Published:** 2023-02-08

**Authors:** Giusi Macaluso, Francesca Grippi, Santina Di Bella, Valeria Blanda, Francesca Gucciardi, Alessandra Torina, Annalisa Guercio, Vincenza Cannella

**Affiliations:** Istituto Zooprofilattico Sperimentale della Sicilia “A. Mirri”, 90129 Palermo, Italy

**Keywords:** *Trypanosoma cruzi*, immunity, toll-like receptors, virulence factors, inflammasome

## Abstract

Chagas disease is a chronic systemic infection transmitted by *Trypanosoma cruzi.* Its life cycle consists of different stages in vector insects and host mammals. *Trypanosoma cruzi* strains cause different clinical manifestations of Chagas disease alongside geographic differences in morbidity and mortality. Natural killer cells provide the cytokine interferon-gamma in the initial phases of *T. cruzi* infection. Phagocytes secrete cytokines that promote inflammation and activation of other cells involved in defence. Dendritic cells, monocytes and macrophages modulate the adaptive immune response, and B lymphocytes activate an effective humoral immune response to *T. cruzi.* This review focuses on the main immune mechanisms acting during *T. cruzi* infection, on the strategies activated by the pathogen against the host cells, on the processes involved in inflammasome and virulence factors and on the new strategies for preventing, controlling and treating this disease.

## 1. Introduction

The infection by *Trypanosoma cruzi* is responsible for a chronic and systemic disease known as Chagas disease that is recognized as a neglected disease by the World Health Organization. The parasite can infect humans and a lot of different species of wild and domestic animals and is mainly transmitted by bloodsucking reduviid insects of the *Triatominae* subfamily through three overlapping cycles: domestic, peridomestic and wild [1]. *Triatoma infestans, Rhodnius prolixus* and *Triatoma dimidiata* are the only competent vectors able to transmit *T. cruzi* to humans. *Triatoma infestans* is mainly spread in the sub-Amazonian endemic regions, *R. prolixus* is reported in South and Central America and *T. dimidiata* is reported in Mexico [2,3].

Other transmission routes to humans involve blood transfusion and vertical transmission [4,5]. In rare cases, ingestion of contaminated foods or liquids or raw meat with a massive parasite infestation can represent an additional transmission route [6]. The life cycle of *T. cruzi* includes several stages in vector insects and host mammals. In insects, the parasite assumes two typical forms identified as replicative epimastigotes and metacyclic tripomastigotes. In mammals, the typical forms are non-replicative blood tripomastigotes and replicative intracellular mastigotes [7]. Various *T. cruzi* strains circulate in mammalian hosts and in insect vectors. This heterogeneity can be responsible for the different clinical manifestations of Chagas disease as well as the differences in morbidity and mortality reported in different geographic areas [8,9].

In humans, Chagas disease usually evolves through an acute phase that can last up to two months, followed by an asymptomatic phase, also called an intermediate or indeterminate phase, and finally, a chronic phase. In most cases, patients affected by *T. cruzi* are asymptomatic or show mild symptoms. However, 10–30% of infected individuals show non-specific symptoms after an incubation period of 5–40 days. These symptoms include abdominal pain; anorexia; fever; malaise; lymphadenopathy; enlarged liver, spleen and lymphnodes; and localized or generalized subcutaneous oedema. In the case of vector transmission, *T. cruzi* inoculation can lead to the appearance of two typical clinical signs at the portal of entry [10]. One of them is the chagoma, a skin rash and oedema at the inoculation site, persisting for several weeks. The other is the Romaña sign which occurs after the accidental deposition of contaminated stools in the conjunctival sac due to eye rubbing. The transfer of tripomastigotes to the conjunctiva causes eyelid edema and conjunctivitis, which is often also associated with lymphadenitis or cellulitis. During the weeks following the bite, some patients may also develop a diffuse morbilliform rash [11,12]. In the case of oral transmission, patients may show nausea and vomiting, diarrhea, jaundice, abdominal pain and gastrointestinal bleeding. In 5–10% of cases, patients with an acute infection can die of myocarditis, encephalitis or meningoencephalitis, with most deaths usually occuring in children [13]. In pregnancy, the infection can lead to premature birth, low birth weight, low Apgar index, hypotonicity, fever, hepatosplenomegaly and anemia [14,15]. Intrauterine infections are also associated with abortion and placentitis [10]. The acute phase is followed by the intermediate phase, which can persist for decades or a lifetime in an asymptomatic form characterized by serological positivity to *T. cruzi* [16]. After several decades, about 30–40% of healthy carriers can develop chronic Chagas disease, showing some organ dysfunctions that can lead to heart failure, congestive and/or gastrointestinal disorders or death [17]. Immunocompromised patients (HIV-infected or organ transplanted) may experience a reactivation of the infection [8,9] (Figure 1). 

The pathogenesis of Chagas disease is not well understood. A long period of parasite persistence can lead to both direct and indirect injures. Direct injuries consist of *T. cruzi*-mediated cellular and neuronal damage, while indirect injuries are caused by the immune response and autoantigens [18,19].

The course of the disease is determined by the balance between the immune response, the inflammatory response of the host tissues and the replicative activity of the parasites [4,20]. Therefore, an ineffective immunological response will increase the size and persistence of a parasitic load and consequently lead to an excessive inflammatory response that causes tissue damage. Conversely, in case of an effective immune response, the parasitic load as well as the inflammatory consequences are minimized, which results in less tissue damage [20]. Therefore, the study of immune mechanisms and parasite activity is necessary to understand the processes leading to the onset of this disease.

The aim of this review is to summarize the current comprehension of the immunological processes activated by *T. cruzi* and of the strategies used by the pathogen to invade the host cells. This information will contribute to the improvement and optimization of new strategies for preventing, controlling and treating Chagas disease.

## 2. The Innate Immune Response to *Trypanosoma cruzi*

Innate immunity, consisting of phagocytes, especially macrophages, neutrophils and dendritic cells, constitutes the first line of defence that *T. cruzi* faces when invading a vertebrate host [21,22,23]. The role of these cells is to recognize pathogen-associated molecular patterns (PAMPs) and damage-associated molecular patterns (DAMPs) by membrane receptors such as toll-like receptors (TLRs). *Trypanosoma cruzi* can be recognized by TLRs and the cytokine production that is subsequently activated by these receptors has an important role in the host’s defence. Nucleotide-binding oligomerization domain-like receptors (NLRs) can also recognize *T. cruzi*; these receptors recognize PAMPs, which can be phagocytosed or can enter into cells through pores, and DAMPs [24]. Furthermore, phagocytes secrete cytokines involved in the promotion of inflammation and in the activation of other defence cells at the infection site. *Trypanosoma cruzi* antigens act as recognition signals and regulate the expression of proinflammatory cytokines from macrophages such as IL-1, IL-12, TNF-α and IL-10. Following *T. cruzi* infection, proinflammatory cytokines and IFN-γ produced by natural killer (NK) cells, together with effector T cells, influence macrophage activation status [25]. IFN-γ can activate macrophages through the classical pathway, even in combination with TNF-α. Activated macrophages produce microbicidal intermediates and reactive oxygen species that can kill *T. cruzi*, triggering a type I polarized response [26]. Instead, IL-4 and IL-13 cytokines released in type II responses induce macrophage activation in an alternative way that counteracts the classical pathway [27,28]. Several parasite antigens induce the classic activation of macrophages leading to NO increase: glycophosphatidylinositol-anchored mucin-like glycoproteins (GPIs) are able to induce IFN-γ-stimulated macrophages to produce NO [29]. The GPI receptor TLR2 acts by stimulating IL-12, TNF-α and NO production [30]. In the presence of PAMPs and DAMPs, dendritic cells (DC), such as macrophages and neutrophils, produce costimulatory molecules and cytokines which allow T cell activation, in combination with the antigen itself [31]. 

DCs and macrophages, which act as professional antigen-presenting cells, are central in the development of immunity or tolerance. The profile of the produced cytokines (TNF-α, IFN-γ, IL-12, IL-22, IL-6 and IL-10) may be different and will depend upon the *T. cruzi* strain [32]. The functions of dendritic cells are affected by *T. cruzi* secreted factors that can induce tolerance by inhibiting TNF-*α* and IL-12 production [33]. 

NK cells are crucial in the innate response because they produce IFN-γ and TNF-α. These cytokines determine parasite elimination by inducing macrophage activation in the early *T. cruzi* infection phase. Activated NK cells (CD16+ and CD56–) were found to be increased in children in the acute phase of the disease [34]. The NK cells form cell–pathogen contacts resulting in a reduction in pathogen motility and an increase in cell membrane damage, leading to the direct elimination of extracellular parasites. This NK-mediated killing action is induced by IL-12 and causes the exocytosis of cytotoxic granules and damage to the parasite cell membrane [23].

Another important component of innate immunity is the complement cascade system, which consists of several plasma proteins that are capable of opsonizing pathogens, thereby recruiting phagocytic cells to the infection point and destroying the infectious agent [23]. The complement cascade functions as a proteolytic enzyme cascade that amplifies signals generated by the pathogen infection and determines parasite elimination. Different pathways (classical, alternative and lectin) can lead to complement cascade activation, converging towards the cleavage of C3 into C3a and C3b [35]. C3a has a proinflammatory action, while C3b is recognized by neutrophil and macrophage receptors that promote pathogen phagocytosis. Moreover, C3b induces the production of the proinflammatory factor C5a [35].

The inflammasome is a multimeric protein complex, which is assembled in the cytoplasm of host cells (which are also cells of innate immunity) following various types of stress signals or in presence of microbial molecules [36]. [36]. The inflammasome generally consists of three components: inflammatory caspases such as caspase-1, an adapter molecule such as ASC, and a sensor protein such as NLRP3. The inflammasome induces the secretion of the inflammatory cytokines IL-1β and IL-18 and guides the host cell towards specific programs of programmed inflammatory death called pyroptosis. These events favor pathogen elimination in the infected and damaged tissues, induce adaptive immunity and return tissue homeostasis. Zamboni and Lima-Junior describe the activation of the NLRP3 inflammasome in response to *T. cruzi* infection. The recognition pattern of *T. cruzi* occurs through TLRs, which are activated following their interaction with pathogenic molecules, and leads to NOD1 activation. This process culminates in the assembly and activation of the NLRP3 / ASC / caspase-1 inflammasome complex, which determines the cleavage of caspase-1 and the release process from pro-IL-1β to the mature cytokine. The authors hypothesize the existence of an additional pathway of ASC-dependent caspase-1 activation that is independent of NLRP3 [36].

## 3. The Adaptative Immune Response to *Trypanosoma cruzi*

The onset of adaptive immunity is followed by the enhancement of circulating activated B lymphocytes that produce and secrete antibodies which play a crucial role in the adaptive humoral immune response. Kumar and Tarleton demonstrated that mice unable to produce antibodies could not control *T. cruzi* growth and died during the acute phase of the disease, thereby demonstrating the importance of the humoral immune response in controlling *T. cruzi* infection [37].

Although B lymphocytes mount an effective immune response to *T. cruzi* during its early stages [20,38], antibodies mostly produced against *T. cruzi* surface antigens may not completely resolve the infection and allow the parasite to permanently infect the host [23]. Cytokines largely coordinate both humoral and cellular immune responses to *T. cruzi* infection. 

Beyond their role as antibody producers, B cells can secrete cytokines, including IL-17 and IL-10, and are responsible for presenting antigens to immune cells, thereby serving as a link between innate and adaptive immunity [23]. 

B cells are fundamental in activating Th1 cell activities which favor the control of parasite growth [20]. A reduction of proinflammatory cytokines (IFN-γ, IL-12) has been shown in spleen supernatants from mice lacking mature B cells [20]; the immune system is unable to differentiate effector CD8+ T cells and to educate a Th1 functional model of T cell cytokines in the absence of mature B cells.

T lymphocytes are crucial in the adaptive cellular immune response. Following the recognition of signals from T cell receptors on the surface of antigen-presenting cells (APCs), the T cell response is activated and naive T (Tn) cells undergo clonal expansion and change the molecular expression and cytokine production, thereby generating T cells with different roles [23]. The T cell activation process results in the production of effector T cells (TE) and also generates memory T cells that are capable of self-renewal and long-term persistence. The differentiation and expansion polarized towards IFN-γ of CD4+ and CD8+ T cells are induced by IL-12, which is produced by DC and NK cells, and trigger CD8+ T cells cytotoxic activity and macrophage effector mechanisms. CD4+ TE lymphocytes stimulate the proliferation of B lymphocytes and the production of antibodies that can determine the lysis of trypomastigotes. Moreover, in the acute phase of the infection, T cells are recruited to the tissues where IFN-γ induces the production of chemokines. A proper balance between inflammatory and anti-inflammatory cytokines and an adequate cellular response must be achieved to avoid tissue damage and keep parasite levels in check [39] (Figure 2).

An increase in activated T lymphocytes and the consequent release of pro- and anti-inflammatory cytokines has been observed in chronic Chagas disease. Most patients with the chronic form of the disease remain asymptomatic; in them a balance is achieved between host and parasite [40]. The chronic form of the disease has been correlated with an increased production of regulatory cytokines (IL-10) compared with inflammatory cytokines (IFN-γ and TNF-α) [41,42]. Therefore, the capacity to produce IL-10 later in the acute phase may have a crucial role in the response control allowing the disease to become chronic.

## 4. Toll-like Receptors

Toll-like receptors (TLRs) are a family of pattern recognition receptors, which are shared by macrophages and other cells involved in innate immunity. TLRs act in the first stages of the immune response by recognizing different microbial structures/patterns [43,44,45].

Specific biological responses are elicited by TLRs via Toll/interleukin-1 (IL-1) receptor (TIR) domain-containing adaptor molecules, including MyD88, TRIF, TIRAP (Mal) and TRAM [46].

These receptors are involved in *T. cruzi* elimination and in phagocyte recruitment at the infection site [47,48,49]. However, the inappropriate activation of these receptors may be related to the establishment of a pathological condition [50,51].

The first studies on TLR activity in *T. cruzi* infection showed a role of TLR2 in mediating the immune system response [52]. Several additional pieces of evidence have been reported since then [46]. 

Multiple TLR ligands of *T. cruzi* are able to activate the innate immune system response and, subsequently, the adaptative immunity response. The latter has been related to protection from the infection but also to pathogenesis [53]. In particular, it was reported that TLR2 activates the small guanine phosphonucleotide-binding protein Rab-5, which induces *T. cruzi* internalization by macrophages [54]. Moreover, when stimulated before the infection, TLR2 is able to promote the survival of infected mice [46,52]. TLR2 and TLR4 are able to sense glycoinositolphospholipids (GIPLs) and GPI anchors present on the trypomastigote cell surface [30,31,32,33,34,35,36,37,38,39,40,41,42,43,44,45,46,47,48,49,50,51,52,53,54,55,56,57], thereby inducing the activation of mitogen-activated protein kinase (MAPK) cascade and nuclear factor-kappa B (NF-κB) pathways. The result is NO and pro-inflammatory cytokine production and the activation of a Th1-type response [58]. Moreover, TLR-9 binds unmethylated CpG motifs present in *T. cruzi* DNA [59] and subsequently induces cytokine production in professional antigen-presenting cells via activation of a Th1 response [13,60].

A triple defect in TLR3, 7 and 9 resulted in mice that were more susceptible to *T. cruzi* infection, as evidenced by higher rates of parasitemia and mortality [61].

Human neutrophils stimulated by *T. cruzi* generate neutrophil extracellular traps (NETs), which are fibrous traps of DNA, histones, and granules that are involved in pathogen killing. A study reported that NET release was reduced as a result of treatment with antibodies against TLR2 and TLR4 [62].

*T. cruzi*-infected macrophages are able to produce higher amounts of extracellular vesicles (EVs) with respect to non-infected cells. These EVs interact with TLR2 and induce translocation of NF-κB to the nucleus, thereby activating the production of pro-inflammatory cytokines (TNF-α, IL-6 and IL-1β) which are able to maintain the inflammatory response [63].

The occurrence of particular TLR4 variants (TLR4 Asp/Gly-Thr/Ile genotype, 299/399 TLR4 haplotype and 299/399 TLR4 haplotype) has been associated with a higher risk of chronicity and severity (cardiac involvement) in oral transmitted Chagas disease [64]. TLR4 agonists reduced parasite burdens in the hearts of *T. cruzi*-infected BALB/c mice, however, they were not able to prevent cardiac damage [65].

Lower levels of TLR2, TLR4, TLR9, TRIF and Myd88 transcripts was associated with the infection of mice with high virulent *T. cruzi* strains. Consequently, reduced IL-12 levels were observed in these mice and led to high parasitemia, myocarditis and mortality [66].

Another study evaluated the association between innate immune receptors, adapter molecules and cytokines and clinical manifestations in patients with different forms of chronic Chagas disease. The authors found an increased expression of TLR8 and IFN-β in digestive and cardiodigestive patients and an increased expression of TLR2, IL-12 and TNF-α in cardiac and cardiodigestive patients [67].

TLR2 inhibition increases the histopathological damage induced by parasites, reduces IL-6 and IL-10 secretion and the expression of proliferation and differentiation markers, while increasing the expression of cell death markers [68].

Galectin-3, a β-galactoside-binding lectin, acting in several biological processes [69], allow macrophages and epithelial cells to bind galactosides of membrane debris obtained from the vacuoles that are used to evade the phagolysosomal pathway of the host by some intra-vacuolar pathogens [45]. Galectin-3 is able to favor cellular infiltration in the hearts of mice infected by the pathogen, collagen deposition and cardiac fibrosis. The authors reported that an unbalanced TLR expression on APCs might compromise the immune response in galectin-3-deficient mice in vivo [70].

## 5. Virulence Factors

*Trypanosoma cruzi*, during its different stages, can infect different host cells using several virulence mechanisms: resistance to oxidative damage, humoral immune response evasion and cell invasion [71]. Different virulence factors act in a sequential manner during the different phases of the *T. cruzi* life cycle. Upon infection, metacyclic trypomastigotes (MT) mainly invade local macrophages, fibroblasts and tissues at the site of infection [72]. The antioxidant mechanisms used by *T. cruzi* are crucial for the inactivation of reactive oxygen and nitrogen species released by the host cells at the early stage of the infection [73]. The parasite produces several enzymes, such as peroxidases, that act on different molecules from the cellular oxidative pathway. Glutathione peroxidase TcGPXI (present in the cytosol) deactivates exogenous hydroper-oxides and TcGPXII (present in the endoplasmatic reticulum) inactivates lipid-hydroperoxides [74]. Ascorbate-dependent heme peroxidase TcAPX disables the binding of hydroxyl ions with oxygen in conjunction with the cytosolic tryparedoxin peroxidase TcCPX and mitochondrial TcMPX. *T. cruzi* also has iron superoxide dismutases (FeSOD) that detoxify reactive oxygen species generated in the cytosol, glycosomes and mitochondria [75]. The expression of enzymes of the *T. cruzi* antioxidative network is related with its life cycle. After transforming into bloodstream trypomastigotes (BT), *T. cruzi* is able to resist the humoral immune response and the lytic effects of the complement system [76]. The evasion mechanism is mediated by the surface glycoproteins of *T. cruzi* trypomastigotes, which restrict the activation of the classical and alternative complement pathways [77]. The trypomastigote decay-accelerating factor (T-DAF) is a surface glycoprotein that interferes with the C3 convertase-mediated assembly of the classical and alternative pathways [78]. The complement regulatory protein (CRP) is a surface-anchored glycoprotein expressed only by trypomastigotes, which inhibits the activation pathway of the complement system [77]. T-DAF and CRP are trans-sialidase-like glycoproteins belonging to the *T. cruzi* trans-sialidase superfamily [79]. Both proteins impair C3b formation by interacting with C4b and C3b [35]. Calreticulin (TcCRT) is a surface molecule that interacts with C1q to inhibit the activation of the classical complement pathway [80]. The complement C2 receptor inhibitor trispanning (TcCRIT) factor impairs the activation of complement cascades via both the classical and lectin pathways through the cleavage of the shared C2 factor and impairs the formation of C3 convertase via its interaction with C4 [35]. The proline racemases (PRs) TcPRACA and TcPRACB are secreted and intracellular enzymes, respectively [81]. TcPRACA is a B cell mitogen which initiates the activation of nonspecific polyclonal lymphocytes and is important for *T. cruzi* evasion and persistence [82]. The overexpression of TcPRAC isoforms results in increased parasite differentiation and cell invasion [83]. Tc52 is a secreted protein responsible for suppressing T cell proliferation [84]. It is able to modulate the expression of macrophage cytokines and iNOS and the production of NO [85]. Once they parasites have differentiated into extracellular amastigotes (EA), they start a new cycle of infection and invade new host cells, therefore they require molecules that allow them an efficient cellular invasion that favors adhesion and the activation of signaling cascades [86]. P21 and TcMVK proteins released by EAs favour host cell invasion. P21 rearranges actin filaments of the host cells and induces actin polymerization and phagocytosis [87]. TcMVK is bound to the membrane of the host cells and induces parasite uptake into HeLa cells [88]. *T. cruzi* has developed surface proteins (transialidases, mucins, mucin-associated surface glycoproteins and phospholipases) that allow the adhesion of metacyclic trypomastigotes and extracellular amastigotes to host cells through interactions with carbohydrates [86]. Gp82 is a surface protein of the metacyclic phase of *T. cruzi* that is responsible for adhesion to the host cell and activation of the Ca2+ signaling cascade, leading to internalization of the parasite [89]. Transialidase enzymes (TS) are important for *T. cruzi* virulence [90] as they allow the pathogen to acquire sialic acid from host cells and modify trypomastigote surface proteins, making them capable of inducing cell paralysis and cell lysis. The host has difficulty activating a neutralizing humoral response because transialidases are excreted in large quantities. The ability of their expulsion in large quantities is associated with increased virulence of the strains [91]. Gp85, present on the surface of BTs, is also a surface glycoprotein of the TS superfamily that participates in cell invasion. This action is mediated by the conserved FLY domain, which is characterized by a tropism for endothelial cells and is able to activate host extracellular signal-regulated kinases [92] and facilitate parasite infection. Mucins are glycoconjugates located on the pathogen surface and can receive sialic acid residues from the host donor by TS. They are classified into two groups: TcMUC present in mammals provides protection against the immune system and accepts sialic acid from TS, which is useful for adhesion, regulation of host immune defense and complement evasion [93], while TcSMUG protects the parasite from the digestive proteases of the insect vector [94]. Mucin-associated surface proteins (MASPs) are a group of proteins mainly found in the infectious pathogen (MT and BT), which favor the invasion [95], survival and multiplication of intracellular amastigotes [96]. Gp35/50 is a mucin-like protein complex that induces the internalization of MTs into the host cell via calcium-mediated pathways [97]. The cysteine endopeptidase cruzipain acts in several processes, such as the degradation of the host tissue, cellular invasion, intracellular development and evasion of the immune response [98]. It is present at major life cycle stages of *T. cruzi* in lysosome-associated organelles and on the membrane in amastigotes [99]. The gp63 proteases (gp63-I and gp63-II) may be involved in evasion of the host immune system [100]. The metalloprotease activity and membrane attachment are known for gp63-I.

## 6. Authophagy

Autophagy is involved in both *T. cruzi* differentiation processes and in the interaction between the parasite and the host cell. Pathogenic protists such as *T. cruzi* can use their own autophagy mechanisms or use the autophagy mechanisms of the host cells in order to establish and maintain the infection in the host. 

Two main models were described to explain *T. cruzi* internalization mechanisms. The first model in based on exocytosis and the second on endocytosis. The latest data have shown that both models occur in sequence during the invasion of the host cell [101,102,103,104]. According to the first model, parasites activate lysosomal exocytosis, eliciting a cascade of Ca^2+^ signals in the host cells, then enter and form the parasitophorous vacuole (TcPV) with lysosomal characteristics [101,105]. This pathway involves the peripheral pool of lysosomes present in the host cell as well as the microtubules and kinesin which are important for the transport of lysosomes towards the cell membrane [102,103]. According to the second model, the internalization occurs through an invagination of the host cell membrane, generating a TcPV rich in phosphoinositides of the cellular membrane, but not of the lysosomes [106].

It was demonstrated that when this fusion is inhibited, the internalized parasites are unable to remain inside the host cells and end up in the extracellular environment [107,108]. Following the fusion of the lysosome with the TcPV, the parasite loses mobility, and this activates the differentiation of tripomastigotes toward the non-mobile form of amastigotes [109]. This process is induced by the acidic pH of the vacuole [104]. The maturation of the vacuole is also a key process both for the retention of the parasite within the host cell and for the progression of its life cycle.

Autophagy can be modulated by anti-parasitic drugs in order to block the survival of *T. cruzi* in the host [110,111]. Parasitic autophagy is required during the interconversion between epimastogotes, tripomastigotes and amastigotes. These changes make the parasite able to adapt to host changes during its life cycle, making parasitic autophagy an excellent target for trypanocidal drugs. 

Autophagy is involved in various aspects of innate and adaptive immunity and is a mechanism that is genetically regulated by a class of genes named “Genes related to autophagy” (Genes Atg) that act sequentially during the various stages of autophagosome formation and maturation. 

Duque et al. [112] showed that a parasite can induce autophagy with different mechanisms in primary mammalian cells: by increasing its frequency and by increasing the presence of LC3, a protein involved in the formation of autophagosomes and autolysosomes. Another protein involved in autophagy is the major cysteine proteinase of the parasite Cruzaina (Cz), which is expressed in all developmental stages and in lysosome-like organelles. The highest concentration of Cz is found in reservosomes (pre-lysosomal organelles of epimastigotes). Losinnoa et al. [113] reported that the induction of autophagy enhances Cz deposition in the reservosomes, leading to their maturation into lysosomes.

## 7. Therapeutic Approaches

To date, the therapy for the treatment of the disease is mainly limited to Nifurtimox (NFX). This is a drug with a proven anti-parasitic activity, but it has mutagenic and tumorigenic effects [114]. Bruno et al. [115] demonstrated that stilbenic and terphenyl compounds, such as Nifurtimox, induce both apoptosis and caspase-1 with inflammasome activity in parasitic cells. Studies on macrophages have shown that, among the stilbene compounds, ST18 is the one with the greatest antiparasitic activity. The action of ST18 is based on inducing caspase-1, an enzyme involved in the control of parasitemia, in infected macrophages. Considering its anti-proliferative and pro-apoptosis activities and anti-inflammatory, gastroprotective and hepatoprotective capacities, this stilbene compound was found to be a good candidate against *T. cruzi*.

Another drug currently used in the therapeutic approach for Chagas disease is benznidazole (BZN). Both NFX and BZN are effective for acute infections in some cases, reducing parasitaemia drastically, while in other cases they are ineffective, maintaining high levels of parasitaemia and the risk of an evolution towards the chronic phase [116]. However, their use is limited for different reasons, mainly because of the serious contraindications, adverse effects on the skin (BZN) and gastrointestinal apparatus (NFX) and genotoxic effects in pregnancy [117,118,119]. Moreover, in some countries these drugs are not registered or are expensive, so they are not immediately available for patients. Other antiparasitic drugs, e.g., posaconazole, albaconazole, amiodarone, Tak-187 and K777, have been studied with promising in vitro results [120], but their effectiveness in humans is not yet known. Regarding chronic patients, the treatment is focused on their management in a specialized clinical infrastructure, which is very expensive and is often out of reach for patients. Moreover, little is known about the factors influencing the disease progression and about the role of the immune response in parasite reactivation and in the severity of the resulting damage [121,122]. For these reasons, the probability of an optimal treatment for the disease remains uncertain. A therapeutic approach focused only on the control of the parasite load is not sufficient to arrest the progression of chronic disease, rather an immune therapy against parasite persistence, adjuvated with other agents to prevent severe damage, should be the focus of future research [123,124]. In recent decades, research efforts have led to the development of several experimental vaccines, most of them focused on eliciting type 1 T cell-adaptive responses [125], which show promising results in small animal models. 

## 8. Conclusions

Chagas disease mainly occurs in endemic areas of continental Latin America. Due to increased population mobility and migratory flow, the disease has also been recorded in non-endemic countries and is becoming a global health problem [126,127,128,129]. In this review, we highlighted how *T. cruzi* exploits different mechanisms to subvert or evade the host immune response, establishing a complex and dynamic scenario. The review focused on different components of the immune response acting in infection control and on the main mechanisms leading towards the disease progression or to a latency period as well as inducing protective or severe side effects for the host. A more profound analysis of IFN-γ secretion, TLR signaling and macrophage activation as well as the inflammasome pathway is needed to better understand the role of these pathways in immunity against the parasite. Despite recent efforts to clarify the role of the immune system during *T. cruzi* infection, much remains unknown and further studies that take into account the complexity of the disease and the current knowledge of parasite–host interactions are necessary to allow the assessment of possible new immunotherapies for this infectious disease.

## Figures and Tables

**Figure 1 pathogens-12-00282-f001:**
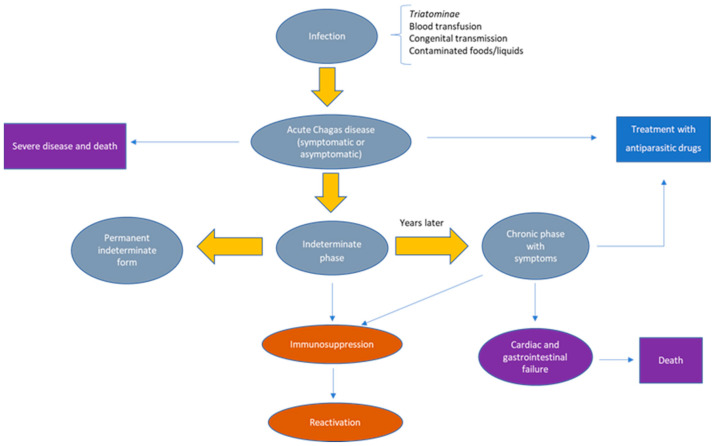
Chagas disease phases and evolution. After exposure to *T. cruzi* infection, the acute phase of Chagas disease is usually asymptomatic or can present as a self-limiting illness. Antiparasitic drug treatment limits the infection and prevents chronic manifestations. Patient death occasionally occurs because of severe myocarditis and/or meningoencephalitis. Some patients presenting the indeterminate form of the disease never develop clinical signs or may show severe and lethal signs of chronic Chagas disease years later. Reactivation of Chagas disease can also occur in chronically infected patients who become immunologically compromised.

**Figure 2 pathogens-12-00282-f002:**
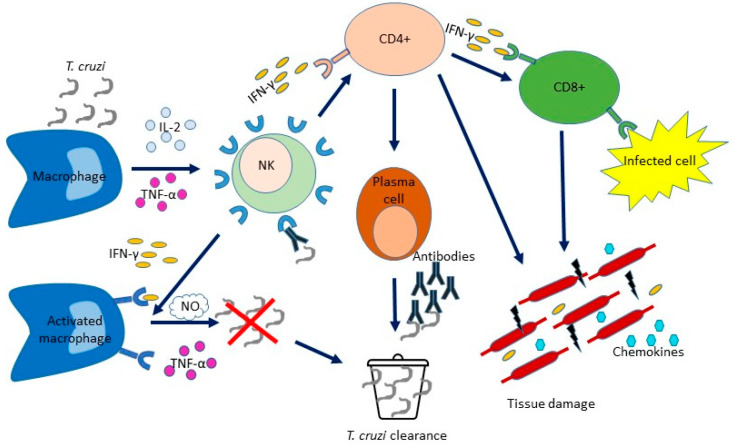
Immune mechanisms acting during *Trypanosoma cruzi* infection. *T. cruzi* infects nucleated cells. The first defence weapon against parasitic infection is mediated by the cells involved in innate immunity (macrophages, DC and NK cells), which occurs before the immune response by specific T and B cells. The production of IL-12 by macrophages is stimulated by *T. cruzi* antigens. IL-12 stimulates NK cells to produce IFN-γ. This, along with TNF-α activity, determines macrophage activation, induces the inflammatory process and controls pathogen replication. The levels of NO produced by macrophages correlate with the control of the parasite load. IL-12 produced by DC and NK cells stimulates the expansion of CD4+ and CD8+ T cells with polarization towards IFN-γ, thereby triggering the cytotoxic activities of CD8+ T cells cytotoxic and the effect activities of macrophages effector. B lymphocytes are stimulated to proliferate and produce antibodies by CD4+ T lymphocytes. During the acute phase, T cells are recruited to the tissues where IFN-γ induces chemokine production. Although the inflammatory environment is critical to the host’s resistance, it can also cause tissue damage.

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
