# Peer review of "A Review on the Immunological Response against Trypanosoma cruzi"

_pathogens, 2023, doi:10.3390/pathogens12020282_

Round 1

Reviewer 1 Report

In this manuscript Giusi Macaluso et al tried to discuss different trajectories of host immune responses in Trypanosoma cruzi infection. The key pathways and associated components have been stated to explain host-pathogen crosstalk. In the current form it is difficult to extract the take home message from the manuscript. The manuscript lacks certain standard to be regarded as a review article for the below shortcomings:

1.       The manuscript has been poorly written.

2.       Lack of coherence is evident.

3.       The projection of the review along with its rationale is unclear in the introduction section.

4.       A proper cascade is absent towards explaining the innate and adaptive immune systems in infection.

5.       A pictorial diagram is pertinent to explain the processes

6.       A table with key components along with their positive and negative roles involved in each pathway is necessary.

7.       The discussion lacks appraisal on future research scopes towards exploring therapeutic and preventive interventions.

Author Response

In this manuscript Giusi Macaluso et al tried to discuss different trajectories of host immune responses in Trypanosoma cruzi infection. The key pathways and associated components have been stated to explain host-pathogen crosstalk. In the current form it is difficult to extract the take home message from the manuscript. The manuscript lacks certain standard to be regarded as a review article for the below shortcomings:

- The manuscript has been poorly written.

- Lack of coherence is evident.

- The projection of the review along with its rationale is unclear in the introduction section.

-  A proper cascade is absent towards explaining the innate and adaptive immune systems in infection.

- A pictorial diagram is pertinent to explain the processes

- A table with key components along with their positive and negative roles involved in each pathway is necessary.

- The discussion lacks appraisal on future research scopes towards exploring therapeutic and preventive interventions.

We thank the reviewer for his/her comment. We have carried out an extensive review of the manuscript, improving it, giving it greater coherence, and making the projection of the review clear up to the introduction section. We better explained innate and adaptive immunity and included a pictorial diagram to better explain the processes as suggested by the reviewer. Finally, we also enriched the conclusions.

Reviewer 2 Report

The article was conducted to review the immune processes included during the T. cruzi infection. This knowledge is useful for the development of new therapies targeting directly the parasite or modulating the host immune system or for optimizing new vaccine strategies. However, given the importance of this subject, the authors should be correct and/or elaborate on the indicated points stated below.

Major Points

- The order of the subheadings should be rearranged.

- T. cruzi lipopolysaccharide section should be explained in detail.

Minor Points

-All text should be revised for typos. Some examples;

Line 37: negletted should be neglected

Line 42: Triatominae should be written in italic

Line 64: non-specific is better to use instead of “aspecific”

Line 66: “Two typical signs that appear in case of vector transmission, are the Chagoma and the Romaña sign” this statement should be expressed differently

-All references must be in standard format

Round 2

Reviewer 1 Report

The manuscript is acceptable in its present form. However, minor language corrections are needed.

Author Response

Dear Editor,

Please find attached the revised version of our manuscript entitled “A review on the Immunological Response against Trypanosoma cruzi”. We revised our manuscript following reviewers’ comments.

We attached a word file with the track changes made to ease your perusal of our manuscript changes.

We hope that this version of the manuscript will be accepted for publication.

Thanks for your consideration,

Best regards

Reviewer 2 Report

All corrections are appropriate.
